# Compromised Chondrocyte Differentiation Capacity in TERC Knockout Mouse Embryonic Stem Cells Derived by Somatic Cell Nuclear Transfer

**DOI:** 10.3390/ijms20051236

**Published:** 2019-03-12

**Authors:** Wei-Fang Chang, Yun-Hsin Wu, Jie Xu, Li-Ying Sung

**Affiliations:** 1Institute of Biotechnology, National Taiwan University, Taipei 106, Taiwan; weifange@gmail.com (W.-F.C.); r05642002@ntu.edu.tw (Y.-H.W.); 2Center for Advanced Models for Translational Sciences and Therapeutics, University of Michigan Medical Center, Ann Arbor, MI 48109, USA; jiex@med.umich.edu; 3Agricultural Biotechnology Research Center, Academia Sinica, Taipei 115, Taiwan; 4Animal Resource Center, National Taiwan University, Taipei 106, Taiwan; 5Center for Biotechnology, National Taiwan University, Taipei 106, Taiwan

**Keywords:** telomeres, telomerase, embryonic stem cells, mesoderm, mouse

## Abstract

Mammalian telomere lengths are primarily regulated by telomerase, consisting of a reverse transcriptase protein (TERT) and an RNA subunit (*TERC*). We previously reported the generation of mouse *Terc*^+/−^ and *Terc*^−/−^ embryonic stem cells (ntESCs) by somatic cell nuclear transfer. In the present work, we investigated the germ layer development competence of *Terc*^−/−^, *Terc*^+/−^ and wild-type (*Terc*^+/+^) ntESCs. The telomere lengths are longest in wild-type but shortest in *Terc*^−/−^ ntESCs, and correlate reversely with the population doubling time. Interestingly, while in vitro embryoid body (EB) differentiation assay reveals EB size difference among ntESCs of different genotypes, the more stringent in vivo teratoma assay demonstrates that *Terc*^−/−^ ntESCs are severely defective in differentiating into the mesodermal lineage cartilage. Consistently, in a directed in vitro chondrocyte differentiation assay, the *Terc*^−/−^ cells failed in forming Collagen II expressing cells. These findings underscore the significance in maintaining proper telomere lengths in stem cells and their derivatives for regenerative medicine.

## 1. Introduction

The ends of eukaryotic chromosomes are capped with telomeres, copies of a hexamer repeat sequence, and associated proteins, which play central roles in stabilizing the ends of chromosomes during replication. Telomeres are primarily regulated by telomerase, a ribonucleoprotein (RNP) consisting of the protein subunit TERT and the RNA subunit *TERC*. A short segment within *TERC* is used as a template for a TERT-catalyzed reverse transcription reaction to elongate telomeres. At the organism level, telomerase is inactivated in somatic cells, whose telomeres shorten each time they proliferate [1,2]. In embryonic tissues and adult stem cells (e.g., hematopoietic stem cells), telomerase is expressed as a mechanism to maintain proper telomere lengths [3,4,5]. In humans, loss of function mutation in genes that encode telomerase components, such as *TERT or TERC*, cause premature aging and age-related diseases, including dyskeratosis congenital (DC), aplastic anemia, and idiopathic pulmonary fibrosis (IPF), which are collectively referred to as “telomere syndromes” to reflect the short and dysfunctional telomeres commonly found in these patients’ cells [6,7,8]. Because mice are characterized with excessive long telomere lengths (~100 kb vs. ~10–15 kb in humans) [9,10], early generations of telomerase deficient mice that are homozygous *Tert* or *Terc* knockout are viable and displayed no obvious ageing phenotypes. At generation three, *Terc*^−/−^ mice were reported to exhibit accelerated age-related bone loss starting at three months of age [11]. Starting at generation five, *Tert*^−/−^ and *Terc*^−/−^ mice displayed premature ageing phenotypes such as grey hair, wrinkled skins, as well as dramatically decreased fertility, indicating that a threshold short telomere length, once it is reached, would perturb normal functions of various types of cells [11,12,13,14,15,16].

Pluripotent stem cells (PSCs), including embryonic stem cells (ESCs), nuclear transfer embryonic stem cells (ntESCs), and induced pluripotent stem cells (iPSCs), represent promising resources for regenerative medicine [17]. The two major characteristics of PSCs are that they possess the capacity for unlimited self-renewal, and that they maintain a pluripotency status, such that they can differentiate into various cell types and in particular all germ layer lineages including endodermal, mesodermal and ectodermal lineages. It is conceivable that telomerase insufficiency in PSCs will compromise the self-renewal and subsequently the differentiation competency [17,18]. Indeed, TERT-knockout human ESCs (hESCs) show telomerase inactivation that results in progressive telomere shortening, apoptosis, or limited proliferation capacity [19,20,21]. In mice, ESCs derived from late generation *Terc*^−/−^ embryos display aneuploidy and chromosomal abnormalities, have compromised differentiation capacity in vitro, and fail to produce any chimeras after blastocyst injection experiments [3].

Consistently, after establishing ESCs (ntESCs) by using tail tip fibroblast cells for somatic cell nuclear transfer (SCNT), we demonstrated that *Terc*^+/−^ ntESCs, whose telomeres were robustly elongated after SCNT and the ESC derivation processes, supported term pup development in the tetraploid complementation assay; whereas *Terc*^−/−^ ntESCs, whose telomeres remained critically short, failed to do so [22]. Our work indicated that telomerase insufficiency-related telomere shortening contributes to defective embryonic development. Previous study also showed that deficiency of telomerase impairs the differentiation of mouse mesenchymal stem cells into adipocytes and chondrocytes [23].

In the present work, we employed the in vitro embryoid body (EB), in vivo teratoma, as well as an in vitro directed differentiation assay to evaluate the differentiation capacities of *Terc*^−/−^, *Terc*^+/−^, and wild-type (*Terc*^+/+^) ntESCs. Our work reveals that the mesodermal lineage chondrocytes differentiation is severely compromised in *Terc*^−/−^ ntESCs.

## 2. Results

### 2.1. Shortened Telomeres and Slowed Cell Growth in Terc^−^/^−^ ntESCs

Mouse ntESCs of different *Terc* genotypes (Figure 1A) with genotype-dependent Telomerase activity (Telomerase Repeated Amplification Protocol (TRAP) assay, (Figure 1B), at Passage 17 to 20, were subjected to semi-quantitative reverse transcription-polymerase chain reaction (RT-PCR) (Figure 1C), western blot (Figure 1D), and immunofluorescence staining (Figure 1E) to evaluate the expression of conventional pluripotent marker genes. Pluripotent gene expression profiles were indistinguishable among *Terc*^−/−^, *Terc*^+/−^ and wild-type *Terc*^+/+^ ntESCs. However, the cell growth rate was *Terc* genotype dependent, with the slowest in *Terc*^−/−^ ntESCs and the fastest in wild-type cells (Figure 1F). The telomere lengths, as measured by the Southern blot (Figure 1G) and T/S ratio (Figure 1H), were also *Terc* dependent; the longest in the wild-type, followed by heterozygous knockout, and the shortest in the homozygous knockout. These results show that the expression of conventional pluripotency markers was not sensitive to telomerase insufficiency, while telomere lengths and the cell grow rate were.

### 2.2. Compromised Spontaneous In Vitro and In Vivo Differentiation Capacity in Terc^−^/^−^ ntESCs

The germ layer differentiation potency of ntESCs of different *Terc* genotypes were first evaluated by spontaneous in vitro differentiation of embryoid bodies (EBs). Although EBs can be derived from all genotypes of ntESCs with similar EB formation efficiency, the size of EBs from *Terc*^−/−^ ntESCs was significantly smaller than those derived from *Terc*^+/+^ and *Terc*^+/−^ ntESCs (Figure 2A,B). RT-PCR assay reveals that these EBs, regardless of genotypes, all expressed signature germ layer-specific genes, including endoderm, SRY (Sex-Determining Region Y)-Box 17 (*Sox17*) and GATA binding protein 4 (*Gata4)*; mesoderm, *Brychury (Bry)* and heart and neural crest derivatives expressed 1 (*Hand1*); and ectoderm, paired box 6 (*Pax6*), at similar levels with the exception of ectodermal marker gene SRY (sex determining region Y)-box 1 (*Sox1*), whose expression level was significantly lower in the *Terc*^−/−^ group than in those in the *Terc*^+/−^ groups (Figure 2C,D).

We next conducted a more stringent pluripotency test, the teratoma assay. ntESCs of 1 × 10^6^ of each genotype were transplanted to the hind legs of the BALC/C Nu mice. The average weight of the teratoma was smaller in the *Terc*^−/−^ group (0.41 ± 0.12 g), as compared to the *Terc*^+/−^ (1.13 ± 0.71 g) and wild-type groups (0.79 ± 0.20 g) (Figure 2E). Furthermore, hematoxylin and eosin (H&E) staining of teratoma in the *Terc*^−/−^ group failed to reveal cartilage, a mesoderm derived cell type that was observed in *Terc*^+/−^ and wild-type groups; whereas cells for endoderm (e.g., ciliated epithelium) and ectoderm (e.g., neuron like) were observed in all groups (Figure 2F). These results indicate that *Terc*^−/−^ ntESCs possess compromised germ layer differentiation capacity.

### 2.3. Compromised In Vitro Differentiation Capacity to Chondrocytes in Terc^−/−^ ntESCs

To confirm the observation in the teratoma assay that mesodermal cartilage differentiation capacity in *Terc*^−/−^ ntESCs was compromised, we subjected ntESCs of different *Terc* genotypes to a 30-day long directed differentiation protocol for derivation of chondrocytes in vitro (Figure 3A). It has been reported that this chondrogenic differentiation protocol leads to the formation of cartilage with its typical extracellular matrix. The expression of collagen type II (*Col2a1*) is used as an indicator for chondrocytes. In addition, the presence of a key molecule within the cartilage matrix, Aggrecan, is used as an indicator for cartilage formation, which is stained dark blue using Alcian Blue.

Along the time course of differentiation, massive cell deaths were observed in the *Terc*^−/−^ group, but not in the wild-type and *Terc*^+/−^ groups (Figure 3A). Furthermore, cells in the *Terc*^−/−^ group did not express either the early cartilage marker SRY (Sex-Determining Region Y)-Box 9 (*Sox9*), or the late cartilage marker *Col2a1* at day 20 or day 30, as determined by semi-quantitative RT-PCR (Figure 3C). In contrast, in the wild-type and *Terc*^+/−^ groups, *Sox9* was detected on both day 20 and day 30, while *Col2a1* was detectable on day 30 but not day 20 following differentiation (Figure 3C,D). Real-time PCR analysis also confirmed the significantly decrease of *Sox9* gene in the *Terc*^−/−^ group compared with the wild-type and *Terc*^+/−^ groups (Figure 3E). Consistently, Alcian blue staining for Aggrecan was positive in cells of wild-type and *Terc*^+/−^ but not *Terc*^−/−^, on day 30 post differentiation (Figure 3B).

These results show that *Terc*^−/−^, but not the wild-type and *Terc*^+/−^ ntESCs, failed to form Collagen II expressing chrondrocytes and Aggrecan-positive cartilages after directed in vitro differentiation.

## 3. Discussion

We discovered that the differentiation capacity of ntESCs to mesodermal chondrocyte is vulnerable to short telomeres associated damages. In an early study, Liu et al. reported similar differentiation defects in mouse *Terc^−/−^* mesenchymal stem cells [23]. These are consistent with clinical findings that telomere syndrome patients suffer primarily in mesoderm lineage cell types, such as muscle, connective tissue, cartilage, bone, and blood cells [24,25,26]. On the other hand, Aguado et al. reported that iPSCs of long telomeres, as indicated by the high expression of the shelterin-complex protein TRF1, differentiate more efficiently into cardiomyocytes (mesodermal linage) than those with relatively short telomeres [6]. These findings indicate that the differentiation capacity of stem cells into mesodermal linage cell types are sensitive to telomere lengths.

The transcripts of *Sox1*, an ectodermal marker gene, were significantly decreased in *Terc^−/−^* ntESCs compared to other genotypes (Figure 2D); however, the teratoma assay did not reveal ectodermal differentiation defects of *Terc^−/−^* ntESCs (Figure 2E). It is possible that the decrease of *Sox1* could be compensated by other Sox family proteins [27,28]. It remains to be further evaluated if and to what extent the ectodermal differentiation is affected in ntESCs of telomerase deficiency.

More generally, our work underscores the importance for quality control of proper telomere maintenance in regenerative medicine. On one hand, telomere length maintenance is critical for the unlimited self-renewal, pluripotency, and chromosomal stability of PSCs [17]. On the other hand, critically short telomeres lead to slowed self-renewal, chromosomal instability, and compromised pluripotency [17]. Therefore, ensuring healthy telomere lengths should be considered an important quality control parameter toward clinical applications of PSCs and their derivatives. We show, in the present study, that conventional PCR and immunostaining-based examinations of pluripotency markers is unable to identify the germ layer differentiation defects in PSCs that are of short telomeres (e.g., *Terc*^−/−^ ntESCs). Our work suggests that stringent tests, such as a teratoma assay, should be conducted to evaluate the PSCs’ quality especially, when they are at high risk to possess short telomeres, for example haploinsufficiency of a telomerase component gene (*Terc* or *Tert*). This is also valid for trans-differentiation-based regenerative medicine, in which a somatic cell type (e.g., fibroblast) is converted to another somatic cell type (e.g., cardiomyocyte), because during trans-differentiation the somatic cells do not go through a “reprogramming to pluripotency” stage, as in the iPSC derivation process, in which telomeres can be reportedly elongated [29,30,31]. Hence it is very possible that the short telomeres in the original somatic cell type are inherited in the target cell type. Without proper telomere lengths, which are often not evaluated, trans-differentiation derived cells may be suboptimal for the intended therapies.

In summary, the present work demonstrates that *Terc*^−/−^ ntESCs suffer from compromised mesodermal chondrocyte differentiation in vitro and in vivo. It provides new evidence to include telomere length as a parameter in the quality control process in regenerative medicine.

## 4. Materials and Methods

### 4.1. Culturing ntESCs

The generation of wild-type, *Terc*^+/−^ and *Terc*^−/−^ ntESCs was described previously [22]. Briefly, heterozygous telomerase-deficient (*Terc*^+/−^) mice C57B6.Cg-*Terc*^tm1Rdp^/J were purchased from the Jackson Laboratory (Bar Harbor, ME, USA), housed, and bred for generation of *Terc*^−/−^ mice. The tail-tip fibroblast from wild-type C57BL/6, *Terc*^+/−^ and *Terc*^−/−^ generation 2 (G2) mice [3,32] was used as the donor cells for somatic cell nuclear transfer (SCNT) and generated for ntESCs in this study. Animal maintenance, care and procedures described within were reviewed and approved by the Institutional Animal Care and Use Committee of National Taiwan University (NTU) according to protocol number NTU-102-EL-111 (Valid from 2014/08/01 to 2017/07/31). All methods in the manuscript were performed in accordance with the relevant guidelines and regulations of NTU. Maintenance of ntESCs was as described previously [33]. Briefly, cells were cultured in regular embryonic stem cell (ESC) medium, which is composed of Knockout Serum Replacement (KSR, 10828-028, Thermo Fisher Scientific Inc., Waltham, MA, USA), 2 mM Glutamax (2 mM, 35050-061, Thermo), 0.1mM 2-mercaptoethanol (BME, ES-007-E, Merck Millipore, Burlington, MA, USA), 0.1 mM nonessential amino acids (NEAA, 11140-050, Thermo), Penicillin/Streptomycin Solution (P/S, 15140-122, Thermo) in KnockOut DMEM (10829-018, Thermo), and supplemented with 10^3^ unit mouse leukemia inhibitory factor (LIF, ESG1107, Millipore). The E13.5 mouse embryonic fibroblast (MEF) treated with mitomycin C (2 μg/mL, M4287, Sigma, St. Louis, MO, USA) was used as feeders for ntESC culture and passaged every 3–4 days.

### 4.2. In Vitro Differentiation

For spontaneous differentiation of ntESCs, the confluent cells were trypsinized, and aggregated embryoid body (EB) was placed onto the top of a petri-dish by suspension culture, with the concentration of 1000 cells per drop in the differentiation medium, which was composed of GlutaMax, NEAA, BME, P/S in Dulbecco’s Modified Eagle Medium (DMEM, 11995-065, Thermo), and 20% fetal bovine serum (FBS, 10437028, Thermo) in DMEM medium. After 2 days of culture, EBs was re-plated into a new petri dish for further culture for another 8 days, and medium was changed every 2 days. For chondrogenic cells induction, the ntESCs were trypsinized and transferred to the differentiation medium, separated to drops that were 20 μL per drop, and were dripped onto the top of the petri dish. After 3 days, EBs were collected to petri dishes and cultured with the same medium with additional supplementation of transforming growth factor beta 1 (TGF-β_1__,_ 10 ng/mL, , 100-21, PeproTech, Inc., Rocky Hill, NJ, USA), and bone morphogenetic protein-2 (BMP-2, 10 ng/mL, 355-BM, R&D Systems, Minneapolis, MN, USA). EBs were transferred to the tissue culture dishes on gelatin in the differentiation medium containing BMP-2 (10 ng/mL), insulin-Transferrin-Selenium (1 µg/mL, 41400-045, Thermo), and L-Ascorbic acid (50 mg/mL, A4403, Sigma) on the fifth day in chondrocyte differentiation. The medium was changed every 2 days until the thirtieth day in chondrocyte differentiation [34]. Differentiated chondrogenic cells and teratoma section were stained with Alcian blue (A5268, Sigma) for 30 min.

### 4.3. Gene Analysis

RNAs of tissues or cells were extracted by Trizol reagent and reverse-transcribed with SuperScript III First-Strand Synthesis System (18080-051, Thermo) into complementary DNA (cDNA) according to the manufacturer’s instructions for further semi-quantification PCR analysis and real-time PCR analysis. Pluripotent genes were listed as below: *Oct4: 5′*-CTGAGGGCCAGGCAGGAGCACGAG-3′ and *5′*-CTGTAGGGAGGGCTTCGGGCACTT-3′. *Sox2: 5′*-TAGAGCTAGACTCCGGGCGAT-3′ and *5′*-TTGCCTTAAACAAGACCACGA-3′. *Nanog: 5′*-CTTAGAAGCGTGGGTCTTGG-3′ and *5′*-GACTCCAAGGACAAGCAAGC-3′. Three germlayer marker genes were listed as below: *Gata4*: *5′*-TTTCTGGGAAACTGGAGCTG-3′ and *5′*-TGCTTTCTGCCTGCTACACA-3′. *Sox17*: *5′*-CACAACGCAGAGCTAAGCAA-3′ and *5′*-ACTTGTAGTTGGGGTGGTCCT-3′. *Brychruy* (*Bry*): *5′*-ACTGGTCTAGCCTCGGAGTG-3′ and *5′*-TTGCTCACAGACCAGAGACT-3′. *Hand1: 5′*-GCTACGCACATCATCACCAT-3′ and *5′*-GATCTTGGAGAGCTTGGTGT-3′. *Pax6*: *5′*-CCATCTTGCTTGGGAAATCCG-3′ and *5′*-GCTTCATCCGAGTCTTCCCGTTAG-3′. *Sox1: 5′*-CCAAGAGACTGCGCGCGCTG-3′ and *5′*-GGGTGCGCCGGGTGTGCGTG-3′. *Gapdh* was used as the internal control: *5′*-CCCTTCATTGACCTCAACTA-3′ and *5′*-CCAAAGTTGTCATGGATGAC-3′. PCR cycles were amplified by C1000 Thermal Cycler (BioRad, Hercules, CA, USA) and run on agarose gel electrophoresis. Quantitative real-time PCR analysis to detect *Sox9* in chondrocytes followed the procedure described in telomere assay (4.7). Twenty ng of cDNA was used as the template and *Gapdh* was used for normalization. Chondrocyte markers: *Sox9*: *5′*-TGGCAGACCAGTACCCGCATCT-3′ and *5′*-TCTTTCTTGTGCTGCACGCGC-3′. *Col2a1: 5′*-CTGCTCATCGCCGCGGTCCTA-3′ and *5′*-AGGGGTACCAGGTTCTCCATC-3′.

### 4.4. Teratoma Assay

BALB/c Nu mice were purchased from BioLASCO Taiwan Company (Taipei, Taiwan) and maintained in individually ventilated cage (IVC) following the approved protocol reviewed by the Institutional Animal Care and Use Committee of NTU according to protocol number NTU-105-EL-164 (Valid from 2017/08/01 to 2020/07/31). The ntESCs were trypsinized and intramuscular injected into the hind leg 1 × 10^6^ per site. After 6 weeks, mice were sacrificed and the tumors were dissociated and fixed in 10% formalin. Fixed samples were embedded and analyzed by hematoxylin and eosin (H&E) stain.

### 4.5. Terminal Restriction Fragment (TRF) Analysis for Telomere Lengths

Isolation of genomic DNA from ntESCs (i.e., *Terc^−/−^, Terc^+/−^,* and *Terc^+/^**^+^*) and digested with the restriction enzymes HinfI and RsaI. Following DNA digestion, the DNA fragments were separated by 0.8% agarose gel in 1×Tris-acetate-EDTA (TAE) buffer, transferred to a nylon membrane, hybridized with telomere-specific digoxigenin (DIG)-labeled hybridization probe, incubated for 3 h at 42 °C. Hybridized membrane was washed twice at 50 °C with 0.2× saline-sodium citrate (SSC) buffer, and incubated with anti-DIG-alkaline phosphatase. Telomere length imaging was detected by enhanced chemiluminescence with the substrate solution and GeneGnome XRQ Chemiluminescence imaging system (SynGene, Cambridge, UK).

### 4.6. Telomerase Activity Measurement

Telomerase activity was measured by an enzyme-linked immunosorbent assay (ELISA) using a commercial kit (S7700, Millipore). About 1 × 10^5^–10^6^ cells from each sample were lysed by 1×3-[(3-cholamidopropyl)dimethylammonio]-1-propanesulfonate (CHAPS) lysis buffer. Three hundred ng of extract protein from each sample was used for PCR following manufacturer’s instructions. Each PCR experiment included positive control (293T cells) and negative control (heated 293T cells). A serial dilution of TSR8 template provided by the manufacture was used to establish the standard curve. The yield of the PCR reaction was determined by measuring the fluorescence in a spectrofluorometer. Each reaction was performed in duplicate.

### 4.7. Quantitative Real-Time PCR for Telomere Assay

Quantitative PCR (real-time PCR or qPCR) was used to measure relative telomere lengths (RTL) of ntESCs, as previously described [35]. Briefly, genomic DNA was extracted from cells using the DNA Isolation Kit (High Pure PCR Template Preparation Kit, 11796828001, Roche, Basel, Switzerland). For each sample, 20 ng of DNA was used in each reaction. PCR reactions were performed on the SYBR Green detection system (KK4603, Kapa Biosystems, Inc., Woburn, MA, USA), using telomeric primers (5′-CGGTTTGTTTGGGTTTGGGTTTGGGTTTGGGTTTGGGTT-3′ and 5′-GGCTTGCCTTACCCTTACCCTTACCCTTACCCTTACCCT-3′). For each PCR reaction, a standard curve was made by serial dilutions of known amounts of mouse genomic DNA. The telomere signal was normalized to the signal from the single-copy gene (36B4: 5′-ACTGGTCTAGGACCCGAGAAG-3′ and 5′-TCAATGGTGCCTCTGGAGATT-3′) to generate a telomere to single-copy gene ratio (T/S ratio) indicative of relative telomere length of the given sample. Each reaction was performed in triplicate.

### 4.8. Western Blot

Cultured cells were trypsinized, washed with ice-cold Dulbecco’s phosphate-buffered saline twice, and lysed in an adequate volume of 1×Radioimmunoprecipitation assay (RIPA) buffer containing proteinase inhibitor. Extracted proteins were ran on a 12% sodium dodecyl sulfate polyacrylamide gel electrophoresis and transferred to polyvinylidene difluoride (PVDF) membrane. The primary antibodies: mouse-anti-OCT4 monoclonal antibody (0.08 µg/mL, C-10, Santa Cruz Biotechnology, Inc., Dallas, TX, USA), rabbit-anti-SOX2 polyclonal antibody (0.34 µg/mL, GTX101507, GeneTex, Hsinchu City, Taiwan), rabbit-anti-NANOG polyclonal antibody (0.04 µg/mL, RCAB002P-F, ReproCell, Beltsville, MD, USA), and mouse-anti-SALL4 monoclonal antibody (0.05 µg/mL, H00057167-M03, Abnova, Taipei, Taiwan) were used as pluripotent markers. Secondary antibody conjugated with horseradish peroxidase (HRP) was incubated and the membrane was applied with T-Pro LumiFast Plus Chemiluminescent Substrate Kit (JT96-K002, T-Pro Biotechnology, New Taipei County, Taiwan). The signals were detected by GeneGnome XRQ Chemiluminescence imaging system.

### 4.9. Immunofluorescent Staining and Confocal Microscope

Cells were seeded on glass slide and fixed with 4% paraformaldehyde when appropriate confluency was reached. After permeabilization and blocking with 2% bovine serum albumin (BSA) at room temperature, primary antibodies, including SALL4 and SOX2, were incubated overnight at 4 °C. Secondary antibodies, Alexa Fluor 488 goat-anti-rabbit IgG (A21429, 4 μg/mL, Therno) and Alexa Fluor 647 goat-anti-mouse IgG, (A28181, 4 μg/mL, Thermo) were used to detect fluorescent signals and observed with confocal microscope (TCS SP5 II, Leica, Wetzlar, Germany).

### 4.10. Genotyping

Genomic DNA extracted from ntESC lines was used as the template for genotyping. Two sets of primers were used to detect *Terc* knockout and wild-type sequence: oIMR1912-5′-CTCGGCACCTAACCCTGAT-3′, oIMR1913-5′-CGCTGACGTTTGTTTTTGAG-3′, oIMR6916-5′-CTTGGGTGGAGAGGCTATTC-3′ and oIMR6917-5′-AGGTGAGATGACAGGAGATC-3′. The PCR products were run on agarose electrophoresis and the PCR products were represented as *Terc* knockout (280 bp), wild-type (150 bp), and two bands (280 bp and 150 bp) in *Terc* heterozygout samples.

### 4.11. Statistical Analysis

The signal intensity of agarose gel electrophoresis was analyzed with Image J software (version 1.47, National Institutes of Health, Bethesda, MD, USA) [36] and was normalized to internal control gene *Gapdh*. The data were statistically analyzed by GraphPad Prism software by one-way ANOVA following Tukey’s test. Standard deviation of the mean (SEM) was shown in all figures unless indicated. Significance was defined as follows: * = *p* < 0.05, ** = *p* < 0.005, *** = *p* < 0.001.

## 5. Conclusions

In this study, we evaluated the differentiation capacities by in vitro embryoid body (EB), in vivo teratoma, as well as an in vitro directed differentiation assay in *Terc*^−/−^, *Terc*^+/−^, and wild-type (*Terc*^+/+^) ntESCs. Our work revealed *Terc*^−/−^ ntESCs suffered from compromised mesodermal chondrocyte differentiation, which elucidated for the significance in maintaining proper telomere lengths in stem cells and their derivatives for regenerative medicine.

## Figures and Tables

**Figure 1 ijms-20-01236-f001:**
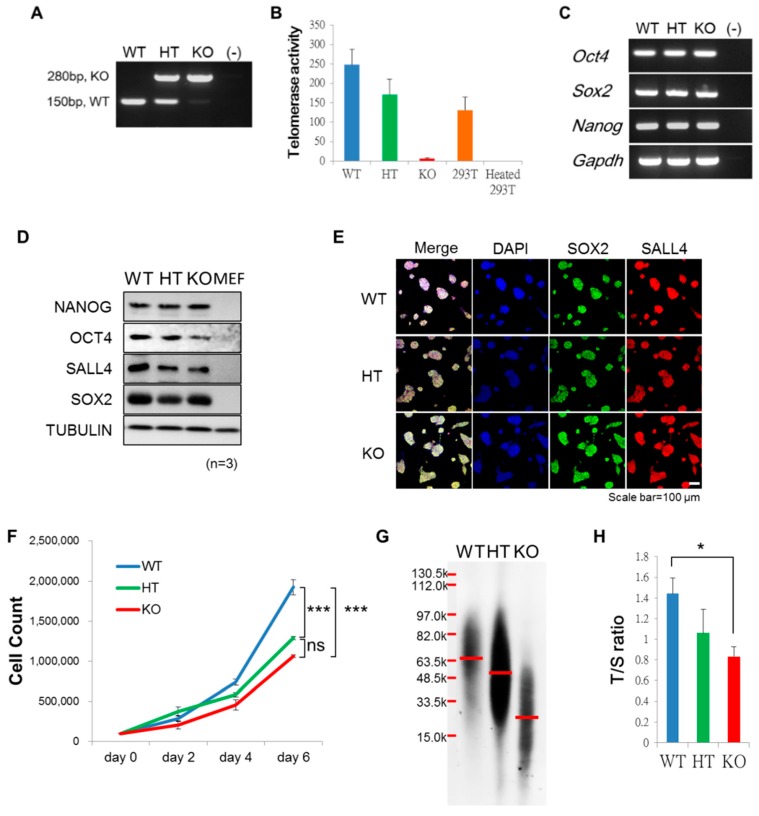
Characterizations of nuclear transfer embryonic stem cells (ntESCs) with different Terc genotypes. (**A**) Genotyping of *Terc*^+/+^ (WT), *Terc*^+/−^ (HT) and *Terc*^−/−^ (KO) ntESCs. (–): negative control without template DNA. The 280 bp band indicates the KO allele and the 150 bp band indicates the wild-type allele. (**B**) Telomerase activity is assayed by Telomerase Repeated Amplification Protocol (TRAP) assay for each *Terc* genotype. Positive control is human 293T cells (293T), and the heat-inactivated 293T cells (Heated 293T) is used as negative control. Error bar indicates the standard deviation (SD). (**C**) Detection of pluripotent markers by semi-quantitative reverse transcription-polymerase chain reaction (RT-PCR). (–): negative control without template cDNA. (**D**) Western blot analysis of pluripotency by detecting NANOG, octamer-binding transcription factor 4 (OCT4), Sal-like protein 4 (SALL4), and SRY (sex determining region Y)-box 2 (SOX2) in ntESC lines. TUBULIN is used as the internal control. Mouse embryonic fibroblast (MEF) is used as the negative control for pluripotent markers. (**E**) Immunofluorescent staining shows the positive signal of SOX2 and SALL4 in ntESC lines. 4′,6-diamidino-2-phenylindole (DAPI) is used for nuclei stain. (**F**) Growth curve of ntESCs in six-days of culture in embryonic stem cell (ESC) medium (*n* = 3). *** indicates significant difference between groups (*p* < 0.001). ns: no significant differences. (**G**) Telomere length analysis by telomere restriction fragment (TRF) in three genotypes of ntESCs. Red line indicates the medium length of genomic DNA. (**H**) Comparison of telomere length by telomere to single-copy gene ratio (T/S ratio). * indicates significant difference between groups (*p* < 0.05) analyzed by unpaired student *t*-test.

**Figure 2 ijms-20-01236-f002:**
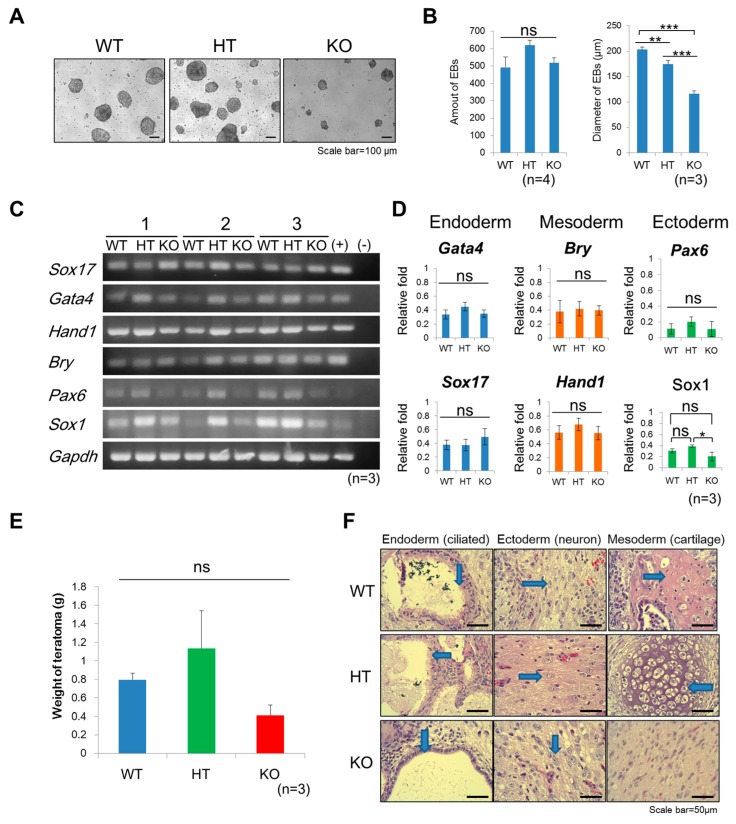
Decreased size of embryoid bodies (EBs) and loss of cartilage cells in teratomas from *Terc^−^*^/−^ ntESCs. (**A**) Morphology of EBs derived from *Terc*^+/+^ (WT), *Terc*^+/−^ (HT), and *Terc*^−/−^ (KO) ntESCs. Spontaneous differentiation is performed in ESC medium without leukemia inhibitory factor (LIF). (**B**) Formation efficiency and diameter of EBs derived from three genotype*s* of ntESC. The size of EBs from *Terc*^−/−^ is significantly smaller than other genotypes (triplicate experiments). ns = no significance. ** = *p* < 0.005, *** = *p* < 0.001. (**C**) Detection of three germ layers markers in EBs. Endoderm markers: SRY (Sex-Determining Region Y)-Box 17 (*Sox17*) and GATA binding protein 4 (*Gata4)*. Mesoderm markers: *Brychury (Bry)* and heart and neural crest derivatives expressed 1 (*Hand1*). Ectoderm markers: paired box 6 (*Pax6*) and SRY (Sex-Determining Region Y)-Box 1 (*Sox1*). Glyceraldehyde-3-phosphate dehydrogenase *(Gapdh)* is used for the internal control. Triplicates of experiments are shown. (+): Positive control, cDNA of the Institute of Cancer Research (ICR) mouse embryo at E7.5 (*n* = 3). (-): negative control without template cDNA (**D**) Quantification of RT-PCR analysis of markers of three germ layers. Error bar indicates the SD, *: *p* < 0.05; ns: no significant differences. (**E**) Size of teratomas from WT, HT, and KO ntESCs is analyzed. Three individual clones of each genotype of ntESCs are assayed as triplicate, ns: no significant differences. (**F**) Teratomas are derived from the WT, HT, and KO ntESCs by injection into immune-deficient Nu mice. Hematoxylin and eosin (H&E) staining of teratomas demonstrated typical cell types of three germ layers, including endoderm (ciliated epithelium), ectoderm (neuron like), and mesoderm (cartilage), indicated by blue arrows. All lineages of cells are observed in *Terc*^+/+^ and *Terc*^+/−^ groups, but loss of cartilage from mesoderm in KO group. Scale bar = 50 μm.

**Figure 3 ijms-20-01236-f003:**
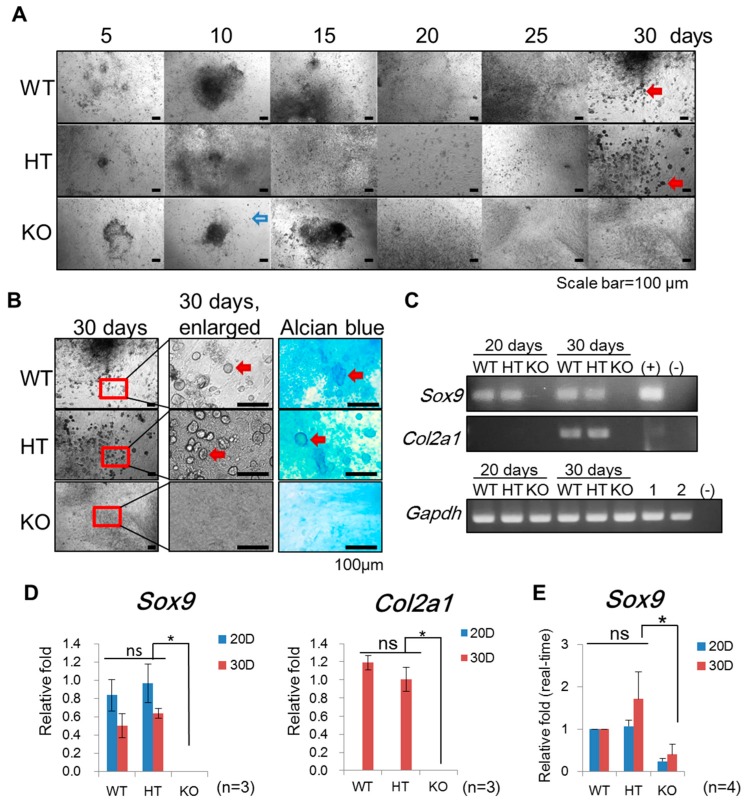
*Terc* depletion abolishes the chondrogenesis in vitro. (**A**) Morphology of chondrocyte differentiation of *Terc*^+/+^ (WT), *Terc*^+/−^ (HT), and *Terc*^−/−^ (KO) ntESCs at different time point. Both *Terc*^+/+^ and *Terc*^+/−^ continues the differentiation procedure till 30 day (red arrows), but the differentiates of *Terc*^−/−^ ntESCs show massive cell death around 10–15 days (blue arrow). Fewer remained cells in *Terc*^−/−^ group. (**B**) After 30 days of differentiation, the cells differentiate into fibroblastic morphology and more chondrocyte-like cells with positive signal stained with Alcian Blue (red arrow indicated). (**C**) Cartilage associated genes are evaluated in differentiated ntESCs. The early cartilage marker SRY (Sex-Determining Region Y)-Box 9 (*Sox9*) and late marker Collagen II (*Col2a1*) expressed at 20 and 30 days of differentiation, indicating the specific induction of mesoderm-chondrocyte from ntESCs. Scale bar = 100 μm. (+): the positive control cDNA for detecting *Sox9* and *Col2a1* collected from mouse testis and cartilage tissue respectively. 1 and 2 indicate the cDNA of testis and cartilage detected with internal control *Gapdh* primer. (-): negative control without template cDNA. (**D**) Quantification of *Sox9* and *Col2a1* expression level at 20 and 30 days (20D and 30D) of differentiation, calculated from the result in (**C**) for triplicate. Error bar indicates the standard deviation. Ns = no significant differences. * = *p* < 0.05. (**E**) Real-time PCR analysis confirms the relative expression level of *Sox9* in differentiated chondrocytes. Cycle threshold (Ct) value for detecting *Sox9* was normalized with *Gapdh* and wild-type at 20 and 30 days, respectively. Ns = no significant differences. * = *p* < 0.05.

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
