# Peer review of "Compromised Chondrocyte Differentiation Capacity in TERC Knockout Mouse Embryonic Stem Cells Derived by Somatic Cell Nuclear Transfer"

_ijms, 2019, doi:10.3390/ijms20051236_

Reviewer 1 Report

The manuscript describes the consequences of altered telomere formation capacity of ntESC of Terc-/- and Terc+/- by using different assays and approaches. The authors analyzed how different Terc mutations: how they effect cell growth, spontaneous differentiation, embryoid body formation and chondrocyte differentiation.

The average quality of the study is appropriate, the results can harbour important information for therapeutic applications and for regenerative medicine.

Suggested minor changes:

1. The introduction section should be supplemented with some more references and data: earlier work of Liu et al, should be mentioned and discussed, which has suggested already, that maintenance of telomere length is an important factor for mesodermal linage differentiation (Telomerase deficiency impairs differentiation of mesenchymal stem cells. Liu L, DiGirolamo CM, Navarro PA, Blasco MA, Keefe DL. Exp Cell Res. 2004 Mar 10;294(1):1-8.). Some relevant information of recent reviews should also be included about telomere structure and function to clarify the importance of Terc more for the readers.

2. The authors refer to their own work (Ref20) about somatic nuclear transfer in the introduction section, but the actual method of producing ntESCs are not given neither in the Materials and Methods section nor as a clear reference in the Results section. The genetic background, clear origin and method of ntESCs derivation used must be given in this publication, as it makes the basis of all further studies.

3. The quality of certain figures (Fig1A.F., Fig3 A,B) could be improved, their size is too small and PCRs should be repeted for 1C Sox2 and Nanog  panels to produce better quality bands (Sox2 can be barely seen and also the band is fuzzy). Overall, the semi-quantitative PRC figures are generally poor quality. QuantitativeRT-PRC method could be used instead of semi-quantitative PRCs to give reliable results.

Author Response

Suggested minor changes:

1. Thank you. We have included new references including Liu et al 2004 in the introduction.
2. Thank you. We have updated the methods part to include this information. The method of genotyping is also provided.       

3. We performed another round of PCR for Fig 1A, 1C to obtain better quality of result images. We also included the Quantitative real-time PCR in our manuscript for Fig. 3E for chondrocyte differentiation for Sox9 gene.

Reviewer 2 Report

This manuscript by Wei-Fang Chang et al reported TERC knockout mouse embryonic stem cells (ESCs) derived from somatic cell nuclear transfer showed compromised mesoderm differentiation.

By utilizing the in vitro embryoid body (EB) differentiation model and in vivo teratoma formation, the authors claimed mesoderm cell fate commitment were impaired in TERC knockout ESCs. Additionally, they induced the ESCs to chondrocyte differentiation to support the notion. However, the evidence provided here is not enough to support the conclusion.  And the interpretation of results were not appropriate organized. This manuscript needs to be revised based on the following concerns:

Major:

1.       The authors exaggerate their finding. They detected only one cell type derived from mesoderm was impaired but draw the conclusion that the mesoderm was compromised in TERC knockout ESCs. The authors should be carefully interpret their result.

2.       Based on the PCR result from Figure 2C and 2D, the ectoderm marker sox1 was decreased in the TERC ESCs but not the mesoderm marker Brachyury and Hand1. This result suggested the ectoderm might be affect but the mesoderm differentiation was similar in TERC knockout ESCs. This result was not consistent with the point of the manuscript. But the authors seemed ignore this result. More analysis should be performed to reveal which germ layer and at which stage were impaired in TERC knockout ESCs.

Minor:

3.       The content in line 23-25 “These findings provide a plausible mechanistic explanation why patients of telomerase insufficiency suffer primarily from damages to mesodermal lineage cell types” should not be included in the abstract section.

4.       The main characteristics of ESCs is unlimited proliferation, self-renew and the pluripotency to differentiate into multiple cell types. The result shown here suggested the proliferation rate of TERC ESCs was decreased. Based on this result, the description in Line 79-80 “…indicating that telomerase activity is dispensable in maintaining ntESCs in vitro” was not correct.

5.       Figure 1A please interpret what are the products at 280bp and 150 bp. And check the label of positive sample group. Why you duplicate Terc+/- ntESCs both as experiment group and positive control.

6.       Check the statistics in figure 1F and 1H. Please keep the mark labeled consistence with other figures through the whole manuscript.

7.        The figure legend 1H “comparism” should be comparison.

8.       The representative images of figure 2A was not match the result shown in 2B. The size of wt ESCs was much smaller than the Terc+/- group.

9.       Positive control in Figure 2C was “organs”. Please indicate which organ was used for each gene.

10.       Line 132-133 “…staining of teratomas demonstrate three germ layers features” should be “demonstrated”.

11.       Please revise the description “Our findings support the current practice of bone marrow (mesodermal linage) transplantation in treating telomere syndrome patients”. The current observation was not relevant to the transplantation.

12.   The genotyping was not mentioned in the materials and methods.

13.   Line 221 should be on “the fifth day”.

14.   Line 222 should be “every two days”

Author Response

1. Thank you. Per Reviewer’s suggestion, we have changed the title to “Compromised chondrocyte differentiation capacity in TERC knockout mouse embryonic stem cells derived by somatic cell nuclear transfer”.

2. Thank you. We agree that this may be an indication that ectodermal differentiation may also be affected. Further work is needed to determine if this is the case and if so to what extent. The results from terotoma assay, that no ectodermal neuron differentiation defects were observed, indicate there may be a compensation mechanism for Sox1. We have included discussion of these observations.

3. This sentence is removed from the abstract.

4. We deleted “indicating that telomerase activity is dispensable in maintaining ntESCs in vitro” in the text.

5. We have redone the genotyping assay and revised the figure image. We also provided description of the 280 bp and 150 bp bands.

6. We performed our statistics by GraphPad Prism software again and the labels of statistics were all updated.

7. Corrected. Thank you.

8. Thank you. The represented images were updated.

9. The positive control should be the cDNA of the ICR mouse embryo at E7.5. This description was added in the figure legend.

10. Corrected. Thank you.

11. This sentence has been removed from the revised manuscript.

12. The method of genotyping was briefly added in the materials and methods “section 4.10”.

13.&14. We have corrected these typos. Thank you.

Round  2

Reviewer 2 Report

The authors has answered all my questions.